# Tuning SERS Signal via Substrate Structuring: Valves of Different Diatom Species with Ultrathin Gold Coating

**DOI:** 10.3390/nano13101594

**Published:** 2023-05-10

**Authors:** Martina Gilic, Mohamed Ghobara, Louisa Reissig

**Affiliations:** Institute of Experimental Physics, Freie Universität Berlin, Arnimallee 14, 14195 Berlin, Germany; mghobara@zedat.fu-berlin.de (M.G.); lreissig@zedat.fu-berlin.de (L.R.)

**Keywords:** diatom valve, SERS, surface-enhanced Raman scattering, guided-mode resonance, finite element method, hybrid SERS sensors

## Abstract

The discovered light modulation capabilities of diatom silicious valves make them an excellent toolkit for photonic devices and applications. In this work, a reproducible surface-enhanced Raman scattering (SERS) enhancement was achieved with hybrid substrates employing diatom silica valves coated with an ultrathin uniform gold film. Three structurally different hybrid substrates, based on the valves of three dissimilar diatom species, have been compared to elucidate the structural contribution to SERS enhancement. The comparative analysis of obtained results showed that substrates containing cylindrical *Aulacoseira* sp. valves achieved the highest enhancement, up to 14-fold. Numerical analysis based on the frequency domain finite element method was carried out to supplement the experimental results. Our results demonstrate that diatom valves of different shapes can enhance the SERS signal, offering a toolbox for SERS-based sensors, where the magnitude of the enhancement depends on valve geometry and ultrastructure.

## 1. Introduction

Since its discovery in 1974 as an outstanding technique for enhancing the Raman signal [1], surface-enhanced Raman scattering (SERS) is gaining increasing interest due to its sensitivity, which could enable single-molecule detection [2]. Despite the exponential growth of publications on SERS, the true mechanism of enhancement is not fully understood, although electromagnetic theory tends to cover all major SERS observations [3]. This technique is based on plasmonically active substrates, which couple laser photons and free electrons within these substrates to induce localized surface plasmon resonances (LSP) and surface plasmon polaritons (SPP) in metallic nanoparticles and the planar metallic surface with adjacent dielectric interfaces, respectively [2]. This leads to an enhancement of electromagnetic fields in close proximity to the surface, where the analyte molecules are adsorbed, consequently enhancing their Raman signal.

In recent years, the sensitivity of the SERS technique has been further improved by employing hybrid substrates, which additionally incorporate dielectric photonic crystals (PCs) or resonant gratings [4]. Such structures have a characteristic interaction with light that can, in a certain wavelength range, obtain a high evanescent field at the surface where the plasmonic structures are often located and thus additionally enhance the SERS signal. For instance, in Hu et al. [5], the presence of a resonant 2D dielectric grating led to the coupling of a guided-mode resonance (GMR) with the LSP, resulting in an improved SERS signal. Nevertheless, the fabrication of such structures often requires clean-room techniques that come with high environmental and financial costs.

As an alternative green solution, some biomaterials have been suggested to replace such artificial dielectric structures in the fabrication of hybrid SERS substrates [6,7,8,9,10]. Among these, diatoms have been proposed as an outstanding source of biosilica with periodic porosity, mainly in the range of visible light wavelengths [11]. Diatoms are unicellular aquatic microalgae enclosed in an intricate porous exoskeleton [12], representing one of the exquisite examples of natural 2D photonic crystals [13,14]. Unlike artificial PCs fabricated with expensive techniques such as lithography, thus being a limiting factor for disposable sensors, diatoms are abundant and versatile [11,14,15]. Diatoms can be cultivated under a wide range of conditions if nutrients and light are provided, and their biosilica can be retrieved [16], modified [15,17], and utilized as building blocks in photonic applications [14,18]. Each of the thousands of species discovered so far has a unique morphology, valve geometry, and pore pattern [12]. Other examples of biomaterials used in fabrication of hybrid SERS substrates include peacock feathers and opal [19], as well as butterfly wings [20].

Diatom biosilica, i.e., mainly their valves, have been successfully employed in the fabrication of various hybrid SERS sensors. These efforts have recently been reviewed in [11]. It has been suggested that diatom valves contribute to SERS enhancement primarily through GMR [11,21] but also by concentrating analyte molecules as well as nanoparticles on their surface and pore rims. In previous work, valves of several diatom species, mainly *Pinnularia* spp., have been utilized to fabricate hybrid substrates for SERS coated with either silver or gold nanoparticles or, less often, thin films [11]. The creation of “hot spots” between nanoparticles (NPs) leads to SERS enhancement factors of up to 10^16^. Nevertheless, homogeneous distribution of NPs remains a big challenge, leaving the hot spots to be placed randomly and thus hampering the reproducibility of the SERS signal. One of the possible strategies to overcome the consequence of the inhomogeneous distribution of NPs could be to create “hot volumes” by placing arrays of weakly coupled metal nanohelices, as in [22]. On the other hand, a continuous metallic thin film may enable homogeneity of the SERS signal across a hybrid substrate. It could also reveal the contribution of structural features of the valve of a given species to SERS and the photonic properties, such as GMR. In the literature, attempts to coat diatoms with homogeneous metallic thin films are scarce [11]. For instance, in Managò et al. [23], *Pseudo-nitzschia multistriata* valves were coated with 20–50 nm thick gold films, where films of 20 nm thickness showed cracks and discontinuities. With increasing film thickness, the uniformity improved, with SERS signal reproducibility reaching 80%. However, the 50 nm film exhibited a roughness of up to 10 nm. Kwon et al. [24] coated *Coscinodiscus* sp. with thin films of 15 nm but without disclosing any data on roughness. Moreover, by using thin plasmonic films, one eliminates the influence of incoming light polarization, because in the case of metallic NPs, the strongest signal is obtained when laser light is polarized parallel to the central axis of the NPs [25], and in the case of rippled surfaces, there is a complex correspondence between patterns of nanogaps, localized hot spots, and polarization [26]. 

In this work, we experimentally evaluate SERS enhancement, homogeneity, and reproducibility obtained with hybrid substrates consisting of biosilica valves of three diatom species—small pennate *Gomphonema parvulum*, medium-sized centric *Aulacoseira* sp., and large centric *Coscinodiscus radiatus*—of very different structural features, coated with an ultrathin uniform gold film of 10 nm thickness. We deliberately chose three extremely distinct species to investigate the contribution of their structural features and the associated optical properties. A numerical analysis using the frequency domain method supplements the experimental results.

## 2. Materials and Methods

### 2.1. Hybrid Substrate Preparation

Three morphologically distinct diatom valves belonging to three different species were scrutinized in this study. Fresh cultures of centric diatom *Coscinodiscus radiatus* (Cosc) and of pennate diatom *Gomphonema parvulum* (Gomp) were kindly provided by Dr. Cathleen Oschatz (Max Planck Institute of Colloids and Interfaces, Potsdam, Germany), while a diatomite (DE) sample was purchased from EP minerals (Reno, NV, USA) with abundant *Aulacoseira* sp. (Aula). The siliceous valves were extracted from the fresh cultures via oxidation using the hot hydrogen peroxide method to remove organic matter, followed by washing several times with deionized water. Fossil valves of *Aulacoseira* sp. (Aula) were purified from the DE sample using the cold HCl method. After purification, the clean valves were kept in deionized water for further use. 

To obtain a monolayer of clean valves, about 60 µL of the valve suspension was spread over clean glass substrates via the drop-casting method and left to dry under ambient conditions. The concentration of the valves was optimized for each sample to avoid agglomeration while forming the monolayer. To obtain a homogenous and well-adhered ultrathin gold film across the valves, a self-assembled monolayer (SAM) was applied to the substrates before the thermal evaporation process. For this, the silica surface of the obtained monolayer was firstly activated using a plasma cleaner (Harrick PDC-32G, Ithaca, NY, USA) in a low-pressure air atmosphere at mid-power, employing RF ≈ 10 MHz for generating the plasma for 10 min. Then, the samples were transferred directly into a vacuum desiccator with an open Eppendorf containing about 80 µL of (3-Mercaptopropyl)trimethoxysilane (MPTMS). Thereafter, the desiccator was evacuated immediately, using a rough vacuum pump for 40 min. After three days in the evacuated desiccator, the substrates were transferred directly to a physical vapor deposition chamber (KJLC Nano 36,Dresden Germany), where a high vacuum (10^−6^ Torr) was applied. The deposition was carried out with an initial rate of 0.1 Å/s and a final thickness of 100 Å (10 nm). The thickness was estimated using an integrated quartz crystal microbalance. After deposition, the hybrid substrates were stored in a clean box for further use. 

In order to compare the hybrid devices with the performance of a bare ultrathin gold film, a reference substrate was fabricated without diatom valves. This created a transparent electrode with a resistance of (45 ± 3) Ω over the whole film. Our previous work showed that this film has a maximum UV–VIS transmittance at 532 nm and a maximum absorbance at 890 nm, where plasmonic resonance reaches the maximum (Appendix A). 

### 2.2. Characterization

Structural characterization of diatom valves as well as of the ultrathin gold layer was performed with a Hitachi SU8030 scanning electron microscope, supported with a secondary electron detector. 

For SERS measurements, about 7 µL of 1 mM Rhodamine 6G (R6G) in ethanol was dropped on top of the gold and left to dry at ambient conditions. SERS measurements and mapping were carried out with a backscattering configuration on a Horiba XploRA confocal Raman instrument equipped with a charge-coupled device (CCD) detector. The spectra acquisition was carried out using an excitation laser wavelength of 638 nm of ca. 40 mW power, in a spectral range of 500–2100 cm^−1^, with an integration time of 15 s per spectrum and averaged over 5 accumulations. Raman mapping was performed with a 1 μm step in the case of Aula and a 0.5 μm step in the cases of Cosc and Gomp, with an integration time of 1 s per step. For focusing the light, the 100x objective (NA = 0.9) was used, giving a beam size of approximately 0.5 μm. Grating was set to 1200 grooves/mm.

### 2.3. Numerical Analysis

The numerical calculations were based on the finite element frequency domain method (FEFD) implemented in COMSOL Multiphysics 5.5. Two-dimensional models representing 2D cross-sections (CSs) across the valves were built based on the structural parameters extracted from SEM analysis. A layer of 10 nm thickness, representing the thin gold film, was added to the CSs. The CSs were analyzed in a large rectangular simulation box, illuminating from the left-hand-side boundary with a transverse electric field of 1 V/m strength. Perfectly matched layers were applied to the other three boundaries. The refractive indices of air and silica were set to 1.00 and 1.46, respectively.

## 3. Results

### 3.1. Morphology and Ultrastructure of the Hybrid Substrates

Scanning electron microscopy (SEM) was performed to reveal the fine structural parameters of diatom valves as well as the homogeneity of the ultrathin gold layer. Our previous AFM measurements confirmed the homogeneity and smoothness of the evaporated gold film on glass activated with the SAM layer, with an average roughness value (Ra) of 0.42 nm (Appendix A). The fine structure of the thin gold film on a valve of *Gomphonema parvulum* is presented in Figure 1c, where the smooth continuous nature of the film without the presence of cracks and voids can be observed. The structure of the gold film on a reference glass substrate without diatom valves as well as over the other two diatom valves (*Coscinodiscus radiatus* and *Aulacoseira* sp.) is presented in Appendix A, with similar observations regarding the quality of the gold film. 

SEM micrographs of the three diatom valves chosen are presented in Figure 1. Gomp valves have an elliptic shape with an approximate length of 7 µm, an average width of 4.6 µm, punctate areolae of 100 nm diameter and 214 nm spacing, and striae spacing of 490–570 nm (Figure 1c). The complete description of Gomp valves is provided in Ghobara et al. [27]. Cosc valves have a circular shape about 100 µm in diameter, with inner pores of 1.2 µm and outer pores of 100 nm diameter, both regularly arranged in hexagonal symmetry (Figure 1a). Unlike Gomp and Cosc, whose valves are of about equal size within the studied samples, cylindrical Aula valves come in a variety of sizes, ranging from 6 to 20 µm valve diameter, and generally follow the rule that the smaller the valve face size, the longer the height. For SERS analysis, we chose the valves with bigger diameters (15–20 µm) that come with a hexagonal arrangement of pores of 250–300 nm in diameter and 0.6–1 µm spacing. In Figure 1b, the top and bottom sides of Aula valves are depicted. While the top side is shaped like a disk with spines at the ridges, the bottom side has a ring-like shape.

### 3.2. SERS Enhancement over Different Substrates

The cationic dye Rhodamine 6G (R6G) was used as a typical probe for SERS measurements. R6G dissolved in ethanol has a strong absorption in the VIS region, with absorption maximum at 554 nm and a vibronic shoulder at 518 nm (Appendix A), which is due to a solvent effect that is slightly red-shifted compared to data from the literature (530 nm) [28]. According to the Franck–Condon rule, the luminescence spectrum was inverted as well as red-shifted [29]. Due to strong luminescence in the VIS region, it is not possible to obtain normal Raman spectra with the typically used green or red laser excitation lines. However, coating the surface with metal should quench the photoluminescence due to non-radiative interaction with the metal surface, allowing the strong SERS signal of R6G to be observed [30]. In our case, the luminescence was still not quenched when using the green laser excitation line at 532 nm on the reference sample R6G on glass coated with gold, limiting the observation of a strong, unhampered SERS signal. In contrast, using the red excitation line at 638 nm on the same sample yields a strong SERS signal with the characteristic modes of R6G [31] at around 610, 770, 1187, 1312, 1360, 1508, 1600, and 1648 cm^−1^. Detailed analysis of the origin of each mode of R6G can be found in [32].

Figure 2a demonstrates the comparison among the SERS spectra of R6G on different hybrid substrates comprising different biosilica valves as well as the reference Au film on glass. Different spectra for each hybrid substrate are shown in the same color, and they stand for the signal reproducibility between different valves. Compared to the reference sample, each of the three substrates dramatically enhances the SERS signal. The averaged signal intensity of each characteristic mode of R6G on the four samples is depicted in Figure 2b, while the relative enhancement of the SERS signal is presented in Figure 2c. Depending on the mode observed, the signal of the spectrum obtained on Cosc was enhanced 6–8 times compared to the SERS signal obtained on the reference sample, while the hybrid substrates based on Gomp showed a 5–7-time enhancement. Finally, the signal of the spectrum obtained on Aula underwent an 8–14-time enhancement, with noticeable variation in the enhancement of different modes. The strongest relative enhancement was observed for the mode at 1312 cm^−1^ (Figure 2c). It must be emphasized that those values do not stand for the absolute enhancement factor of the SERS signal, but instead the relative enhancements compared to the thin gold layer on glass, containing no diatom valves.

Raman mapping was carried out to test the homogeneity on each hybrid substrate, as well as the SERS improvement with respect to the flat unpatterned gold film surrounding the valves. In Figure 3, Raman maps of the intense mode at 1360 cm^−1^ on different hybrid substrates are presented. In the case of the substrate based on Aula valves coated with a thin gold layer (Figure 3a), the mapping is performed with steps of 1 µm. The mapping image corresponds to the optical image, and without a doubt, the signal is homogeneously stronger on the valve than outside of it. As the Cosc valve is much bigger, only a small part of it is mapped with fine steps of 0.5 µm (Figure 3b). Here, in the mapping image, the pore structure is preserved, i.e., the enhancement is stronger inside the pores, suggesting that more light is trapped within. In the case of the small Gomp valve, a mapping with a fine step of 0.5 µm was carried out to image the whole valve as well as the surrounding gold-coated glass substrate (Figure 3c). In this case, the mapping image corresponds to the optical image as well. Nevertheless, the quality of the result is hampered due to a strong luminescence arising from the valve edges.

### 3.3. Numerical Analysis

Numerical calculations were carried out on representative 2D cross-sections of different hybrid substrates to investigate the possible contribution of the valves’ light modulation capabilities, specifically GMR, to the obtained SERS enhancement. The structural parameters necessary to build the 2D CSs, such as pore size and spacing, were extracted from the SEM measurements (see Figure 1). 

The Gomp valve has a distinct 1D grid-like structure, consisting of alternating striae and costae, as demonstrated previously when its GMR was investigated [27]. In Ghobara et al., the longitudinal CSs with grid-like structure were able to couple the light at a specific range of wavelengths inside the grid, leading to the formation of an intense standing wave, with simultaneous enhancement in reflectance and a drop in transmission [27]. By coating the CSs with a 10 nm gold layer, the resonance wavelength maximum λ_GMR_ is slightly shifted, with a reduction in the normalized electric field strength E_Norm,GMR_ within the grid structure. For example, the CS close to the valve edges with maximum striae spacing showed a zero-mode GMR at 610 nm (Figure 4c) instead of 613 nm for the same CS without a gold layer. For all Gomp CSs, no GMR maximum is obtained at the excitation wavelength λ_exc_ = 638 nm. Nevertheless, the field strength of the CS close to the edge is still high in the near field, where the probe R6G molecules are adsorbed in the real measurements (Figure 4c, bottom). 

The created 2D CS of the Cosc valve represents a cross-section through a whole valve, including its mantle (curved edges). After coating the CS with a 10 nm gold layer, a GMR maximum was observed at 700 nm. In Figure 4a, a small part of this CS, showing only two areolae, is presented. At λ_exc_, away from the maximum, the E_Norm_ drops from 3.49 to 2.5 V/m; however, the electric field is still confined inside the areolae (Figure 4a, bottom). This likely matches SERS mapping results, suggesting that light is trapped within the valves regardless of the GMR. 

In the case of the Aula valve, as already described, the structural parameters showed a relatively large variation: pore spacing ranges between 0.6 and 1 μm and pore size between 250 and 300 nm. For the Aula 2D CS with a 10 nm gold layer (Figure 4b), a GMR maximum is obtained at 640 nm (λ_GMR_) for the pore spacing of 0.61 μm, slightly off λ_exc_ (638 nm). At λ_exc_, E_Norm_ drops from 6.27 to 5.88 V/m (Figure 4b, bottom). With increased pore spacing, λ_GMR_ exhibits a strong red shift (Figure 5a), and for the pore spacing of 900 nm, the resonance wavelength appears at 910 nm. It is worth mentioning that at higher λ_GMR_, in the red and infrared part of the spectrum, E_Norm_ drops and consequently spreads more evenly towards the valve edge. In Figure 5a, the pore spacing dependence of λ_GMR_ is demonstrated. Pore size and valve thickness dependence were also simulated (Figure 5b,c) and in both cases, λ_GMR_ as well as E_Norm_ show negligible changes with the simulated parameter.

## 4. Discussion

The exceptional smoothness, adhesion, and conductivity of the ultrathin gold film—in addition to optical transparency—is attributed to the as-used fabrication method involving a molecular adhesive monolayer of MPTMS [33]. Such properties cannot be obtained by evaporating gold directly on a glass substrate [34]. Reducing film roughness on diatom valves helps elucidate the enhancement contribution of the valves with respect to the obtained SERS enhancement. Introducing diatom valves to the substrate adds micro- to nano-scale structuring to the film. On structured metal films, unlike the flat film of the reference substrate, the plasmonic excitations could be localized within nano-scale features, resulting in a combination of propagating and confined plasmon excitations [24]. 

The pore size and pore spacing in diatoms are usually addressed as the key parameters in considering the valves’ photonic crystal features. In many diatom valves, the pore diameter spans four orders of magnitude, from 3 to 2000 nm [12,14,17,35], and the pore spacing is in the same range, in many cases overlapping with the wavelength of visible light. The pore size and spacing of our three structurally distinct valves are comparable with λ_exc_. Pore size and pore spacings of Gomp are smaller than λ_exc_. For Aula, pore sizes are approx. λ_exc_/2, while the spacing matches or exceeds λ_exc_. Finally, for Cosc, the pore size is 2λ_exc_ and the spacing approx. 3λ_exc_. Therefore, the periodic porosity of the valves is similar to PC slabs and could have features such as GMR for off-axis light propagation. Nevertheless, unlike artificial PC slabs, diatom valves are of finite size, with curved edges, and often with more complex symmetries [35]. 

In resonant gratings and PC slabs, with a spacing comparable to the incident wavelength, the GMR can be supported depending on different parameters, including pore spacing, fill factor, thickness, light incident angle, and light polarization [36,37,38]. The supported GMR can couple to plasmonic resonances that can strongly enhance the electromagnetic field at the interface (by evanescence), where the probe molecules are located. This increases the absorption cross-section of the probe molecules, which eventually leads to the enhancement of scattering, including Raman, obtaining a greatly surface-enhanced Raman signal. For this, λ_exc_ should match or be close to λ_GMR_ [21,39,40]. 

The simulation results show that Gomp valves can partially support GMR close to λ_exc_, while the GMR supported by Cosc valves are probably off λ_exc_. In the case of Aula valves, GMR are only supported close to λ_exc_ if the lower value of the pore spacing is considered. However, the single Aula valve has a variation in pore spacing, defects, and imperfections. In the case of defects and imperfections, the porous valve face still supports GMR, but with shifted values, as demonstrated in Appendix A where the central pore is blocked (left) and the cross-section reduced just on the analytical grid (right). The Cosc valves might support a different mechanism by trapping light inside the areolae, acting like a microcavity. Unlike planar PC slabs, the presence of the mantle can couple the light into the valve, as has been suggested through near-field scanning optical microscopy [41]. This can also add up to the observed enhancement in SERS. Finally, it should be noted that the two-dimensional simulations have limitations, especially when considering resonance phenomena, as the valves have three-dimensional structure. The simulations also did not include the influence of the substrate.

## 5. Conclusions

In this work, we fabricated hybrid SERS substrates based on different naturally designed diatom valves coated with a thin smooth gold film. The SERS signal obtained on such hybrid substrates was reproducible and, compared to a reference substrate consisting of a gold layer on glass without diatoms, increased on average by a factor of 6, 7, and 11 in the case of Gomp, Cosc, and Aula, respectively. Our findings suggest that probably all diatom valves can be employed successfully as substrates in SERS-based sensors. Nevertheless, under equal conditions, the magnitude of enhancement varies depending on their geometry and ultrastructure. GMR seems to not be the exclusive mechanism for enhancing the SERS signal in diatom-based SERS hybrid sensors. More efforts are required in the future to understand the real mechanisms for the obtained enhancement. 

## Figures and Tables

**Figure 1 nanomaterials-13-01594-f001:**
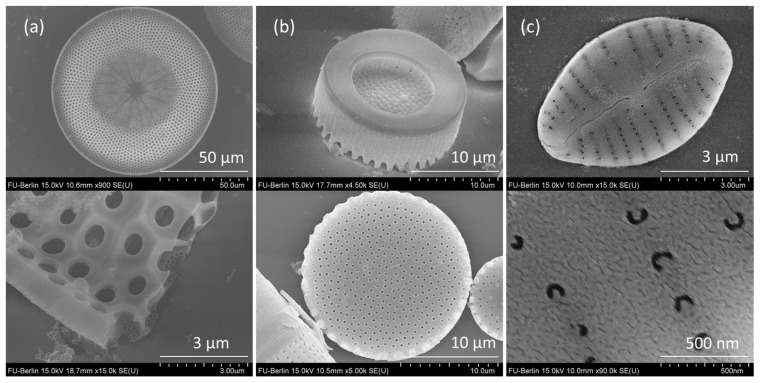
SEM images of (**a**) *Coscinodiscus radiatus*, (**b**) *Aulacoseira* sp., and (**c**) *Gomphonema parvulum* diatom valves with corresponding details.

**Figure 2 nanomaterials-13-01594-f002:**
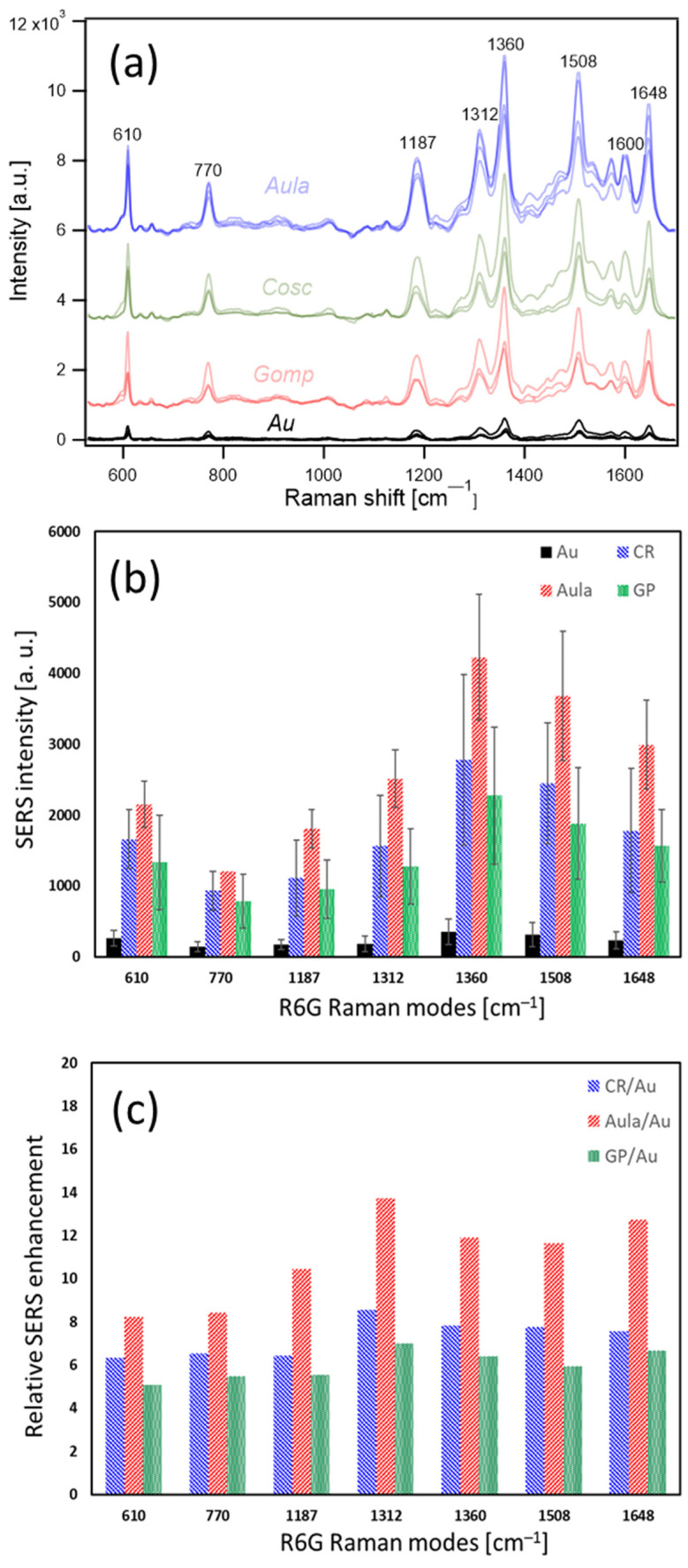
SERS spectra of Rhodamine 6G molecules on different hybrid substrates (**a**), SERS intensity of Rhodamine 6G modes on different hybrid substrates (**b**), and relative SERS enhancement of each mode (**c**).

**Figure 3 nanomaterials-13-01594-f003:**
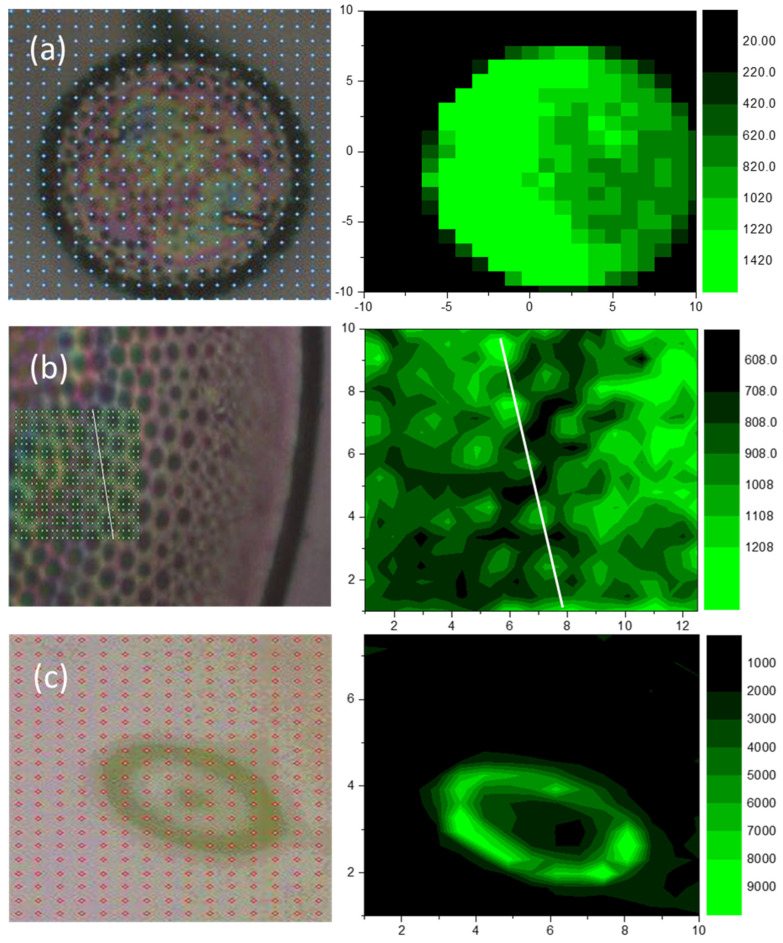
Raman mapping of the intense mode of Rhodamine 6G at 1360 cm^−1^ over Aula (**a**), Cosc (**b**), and Gomp (**c**) single valves covered with thin gold film (**right**) with corresponding light microscope images (**left**).

**Figure 4 nanomaterials-13-01594-f004:**
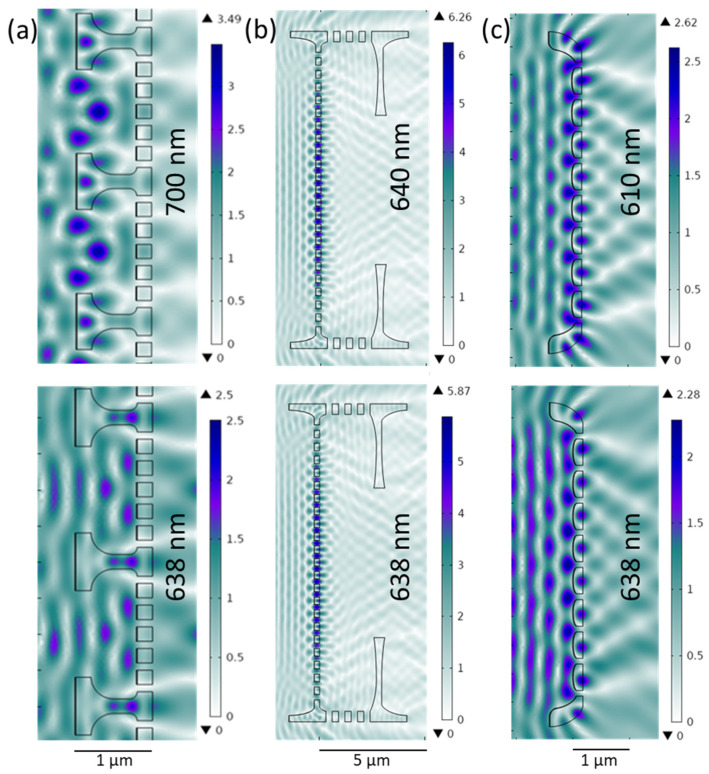
Electromagnetic field enhancement in 2D cross-sections of valve models coated with 10 nm thick gold film at λ_GMR_ (**top**) and λ_exc_ (**bottom**)—detail of the model of Cosc valve (**a**), Aula valve (**b**), and Gomp valve (**c**). The electromagnetic field impinges from the left with an input strength of 1 V/m.

**Figure 5 nanomaterials-13-01594-f005:**
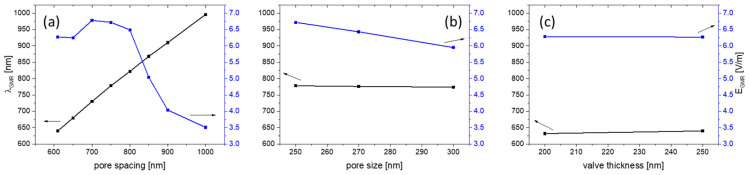
GMR wavelength and E_Norm_ GMR dependence on pore spacing (valve thickness 250 nm, pore size 250 nm) (**a**); on pore size (valve thickness 250 nm, pore spacing 750 nm) (**b**); and on valve thickness (pore size 250 nm, pore spacing 610 nm) (**c**).

## Data Availability

All data needed to evaluate the conclusions in the paper are present in the paper and the Appendix A.

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
