# Peer review of "Tuning SERS Signal via Substrate Structuring: Valves of Different Diatom Species with Ultrathin Gold Coating"

_nanomaterials, 2023, doi:10.3390/nano13101594_

Round 1
Reviewer 1 Report
The paper is devoted to extremely popular investigations of surface enhanced Raman scattering, where enhancing is obtained by regular conducting surface of diatom valves covered with gold - a kind of photonic crystals. Just a few such experiments have been done before, but that photonic crystals were not so regular and hardly reproducible. In this paper special attention was paid to the quality of used valves preparation and obtained coating as well as to Raman intensity measurements and mapping. As a result reproducible enhanced Raman signals are obtained of Rhodamine 6G on three different gold-covered diatom valves, with regular dependence on their geometry and ultrastructure. Reasonable suggestions are proposed about mechanisms of this enhancement on the basis of numerical simulations.
One minor remark: while sample preparations are presented in very detail, any description of Raman experiments is absent completely. No specification of the spectrometer or mapping system, experimental geometry, exciting beam size and power -- just nothing. Pretending to reproducible results such details are necessary.
So the paper deserves publication after this minor revision.
Reviewer 2 Report
The manuscript " Tuning SERS Signal via Substrate Structuring: Valves of Different Diatom Species with Ultrathin Gold Coating" by Gilic et al. shows how SERS signals can be modulated in valves of diatom species coated by gold by changing the structural parameters of the sample.
Furthermore, authors support their claims via electromagnetic simulations. The subject matter of this work is interesting, SERS substrates with homogeneous and enhanced signals are necessary for applications. The presented results are consistent and the manuscript is well-written. I believe this manuscript is suitable for publication if the following issues are addressed:
- Authors should benchmark their results with state-of-the-art substrates in literature in terms of both enhancement, homogeneity and reproducibility. This will ease the reading of the work. It is well know that nanostructured substrates with “hotspots” can largely enhanced near-fields but suffer from a low homogeneity. However, there are already strategies in place able to design optimized nanostructured substrates with large homogeneity by the creation of “hot-volumes” instead (see e.g. [Caridad et al. Scientific Reports, 7, 45548 (2017)]).
- Authors mention in line 103 “For this, the silica surface of the obtained monolayer was firstly activated using a plasma cleaner (Harrick PDC-32G) in a low-pressure air atmosphere at mid-power, employing RF ≈ 10MHz for generating the 105 plasma for 10 min.” ?
The gas used in the process should be stated... argon, oxygen, other.
Reviewer 3 Report
The paper is interesting and well organized. It adds knowledge to manifacturing of SERS substrated topics and nanooptics.
One question is: what do you think the SERS response of the material would be if the coating was done in silver?
I suggest to include ripples based SERS systems by citing: Near-field surface plasmon field enhancement induced by ripples surfaces, D'Acunto etal. Beilstein Journal of Nanotechnology, 2017, 8, 956.
The quality of AFM image is poor because maybe that it has been acquired with a dirty tip, so introducing blurring effects. Can the authors provide other high quality AFM images?
In my opinion, the paper present interestig results and after such changes it could be published.
Round 2
Reviewer 2 Report
Authors have properly addressed all my concerns. I recommend the publication of the manuscript as it is.